# Prognostic value of the systemic immune-inflammation index in lung cancer patients receiving immune checkpoint inhibitors: A meta-analysis

**Yanhui Yang, Ji Li, Yi Wang, Lei Luo, Yi Yao, Xiaoyang Xie** [ID]*

Department of Thoracic Surgery, The First People's Hospital of Neijiang, Neijiang Affiliated Hospital of Chongqing Medical University, Neijiang, Sichuan, P.R. China

* 13990559066@163.com

## Abstract

### Purpose

To explore the association between the systemic immune-inflammation index (SII) score and prognosis in immune checkpoint inhibitor (ICI)-treated patients with lung cancer.

### Methods

PubMed, EMBASE, Web of Science, and CNKI databases were searched up to August 1, 2024. Progression-free survival (PFS) and overall survival (OS) were the primary outcomes queried. Hazard ratios (HRs) and 95% confidence intervals (CIs) were combined, and subgroup analysis was based on pathological type [non-small cell lung cancer (NSCLC) vs. small-cell lung cancer (SCLC)], lines of ICIs (first-line vs. second- or further-line), and combinations of other therapies (yes vs. no).

### Results

Twenty retrospective studies with 2424 participants were included. The pooled results demonstrated that an elevated SII was associated with poorer PFS (HR = 1.82, 95% CI: 1.49–2.21; P < 0.001) and OS (HR = 2.31, 95% CI: 1.73–3.09; P < 0.001) in lung cancer patients receiving ICIs. Subgroup analysis stratified by pathological type, lines of ICIs and combinations of other therapies for PFS and OS further revealed the predictive role of the SII in ICI-treated lung cancer patients.

### Conclusion

Based on current evidence the SII is significantly related to prognosis and could serve as a reliable prognostic indicator in lung cancer patients receiving ICIs.

**Data Availability Statement:** All relevant data are within the paper and its Supporting Information files.

**Funding:** The author(s) received no specific funding for this work.

## Introduction

Lung cancer, including non-small cell lung cancer (NSCLC) and small cell lung cancer (SCLC), is the most common malignancy and leading cause of tumor-related death worldwide [1–3]. Despite great advances in early screening and surgical techniques for lung cancer in recent decades, advanced-stage lung cancer still accounts for a significant proportion of all lung cancer cases [4–6]. Immune checkpoint inhibitors (ICIs) have become one of the most important therapies for advanced lung cancer, especially in driver-negative NSCLC patients [7, 8].

Currently, ICIs include mainly cytotoxic T-lymphocyte-associated antigen 4 (CTLA-4) inhibitors, anti-programmed death-1 (PD-1) inhibitors, and anti-programmed death-ligand 1 (PD-L1) inhibitors. PD-1 regulates T-cell activation by binding to PD-L1 and programmed death ligand 2 (PD-L2). PD-1-generated signaling terminates early TCR signaling by preventing the phosphorylation of key TCR signaling intermediates and reducing T-cell activation and cytokine formation. Therefore, PD-1 inhibitors can interrupt the negative regulatory signals of T cells and ultimately inhibit tumor growth [9]. Compared with standard chemotherapy, the application of PD-1/PD-L1 inhibitors has shown good efficacy in a number of clinical studies, increasing the efficacy and prognosis of patients [10]. Unfortunately, a significant proportion of patients do not benefit from ICIs, even those with a high PD-L1 tumor proportion score [11].

PD-L1 expression, tumor mutational burden (TMB), circulating tumor DNA (ctDNA), and microsatellite instability-high (MSI-H) status are common clinical biomarkers for predicting ICI efficacy [12]. However, the detection cost of these biomarkers is relatively high, and the detection process is complicated, which limits their clinical application. Thus, identifying more economical and convenient biomarkers to predict the therapeutic efficacy of ICIs in clinical practice is necessary.

Growing evidence has indicated that inflammation-, nutrition-, and immune-related peripheral blood indicators play a role in predicting tumor-related immunotherapy efficacy [13–15]. The long-term survival of patients with malignant tumors is closely associated with host inflammation and immunotrophic status [16, 17]. Inflammation is an important feature of the tumor microenvironment and is related to the poor prognosis of patients with tumors [16, 17]. Hematologic inflammatory parameters such as neutrophils, lymphocytes, and platelets can reflect the balance between tumor immunity and inflammation and have a certain predictive effect on the prognosis of patients with tumors [18, 19]. The systemic immune-inflammation index (SII), based on the above blood count parameters plus platelet count * neutrophil count / lymphocyte count, has been reported to have high prognostic value in solid cancer patients treated with ICIs [20, 21]. However, whether this could serve as a reliable prognostic indicator in ICI-treated lung cancer patients remains unclear.

Therefore, we aimed to further identify the prognostic value of the SII among ICI-treated lung cancer patients.

## Materials and methods

The current meta-analysis was performed according to the Preferred Reporting Items for Systematic Reviews and Meta-Analyses 2020 [22].

### Literature search

The PubMed, EMBASE, Web of Science, and CNKI databases were searched from inception to August 1, 2024, for available studies. The following terms were used: PD-1, PD-L1, CTLA-4, ICI, immune checkpoint inhibitor, lung, pulmonary, cancer, tumor, carcinoma, neoplasm,

survival, prognosis, prognostic, systemic immune-inflammation index, and the SII. The specific search strategies were as follows: (PD-1 OR PD-L1 OR CTLA-4 OR ICI OR immune checkpoint inhibitor) AND (lung OR pulmonary) AND (cancer OR tumor OR carcinoma OR neoplasm) AND (survival OR prognosis OR prognostic) AND (systemic immune-inflammation index OR SII). Furthermore, MeSH terms and free texts were applied and all the references cited in the included studies were also reviewed.

### Inclusion criteria

Studies that met the following criteria were included: 1) patients were diagnosed with primary lung cancer pathologically; 2) patients received ICIs with or without the combination of other antitumor therapies; 3) the SII was calculated before immunotherapy as follows: platelet count * neutrophil count / lymphocyte count; 4) patients were divided into elevated and normal-SII groups; and 5) progression-free survival (PFS) or (and) overall survival (OS) were (were) compared between the two groups, representing hazard ratios (HRs) with corresponding 95% confidence intervals (CIs).

### Exclusion criteria

Studies that met the following criteria were excluded: 1) reviews, case reports, editorials, letters, or animal trials; 2) HRs were not directly reported; 3) duplicated or overlapping data; 4) immunotherapy was applied as neoadjuvant immunotherapy; and 5) low-quality studies with a Newcastle–Ottawa Scale (NOS) score $\leq$ 5 [23].

### Data collection

The following information was extracted: first author, publication year, country, sample size, pathological subtype, tumor stage, lines of ICIs, combination of therapy, detailed drugs of ICIs, cutoff values of the SII, endpoint, NOS score, HR, and 95% CI of PFS and OS.

### Methodological quality assessment

The NOS scoring tool was used to evaluate the quality of the included studies. Studies with an NOS score $\geq$6 were included.

Two authors (Yanhui Yang and Ji Li) independently performed the literature search, selection, data collection, and methodological quality assessment, and all disagreements were resolved by team discussion.

### Statistical analysis

All statistical analyses were performed using STATA (version 12.0) software. Heterogeneity between studies was assessed using $I^2$ statistics and the Q test. If significant heterogeneity was detected ($I^2 > 50\%$ and/or $P < 0.1$) the random effects model was applied; otherwise, the fixed effects model was used. HRs and 95% CIs were combined to evaluate the association between the SII and survival. Subgroup analysis based on pathological type (NSCLC and SCLC), lines of ICIs (first-line vs. second-line or further line), and combination of other therapies (yes vs. no) was conducted. Sensitivity analysis was conducted to detect the sources of heterogeneity and assess the stability of the overall results. Furthermore, Begg's funnel plot and Egger's test were conducted to detect publication bias, and significant publication bias was defined as $P < 0.05$ [24, 25].

## Results

### Literature search process

One hundred and seventy-nine records were identified from four databases, and 35 duplicated records were removed. After reviewing the titles and abstracts, 114 records were excluded. Seven studies were excluded because of insufficient data and one that contained duplicated data. Finally, 20 studies were included [26–45]. The detailed process is illustrated in **Fig 1** and specific information for each record was shown in **S1 Table**.

### Basic characteristics of the included studies

All included studies were retrospective and involved a total of 2424 patients. Most studies were conducted in China (14/20) and focused on patients with NSCLC (16/20). Moreover, most enrolled patients had advanced-stage disease (TNM III–IV or extensive stage). The SII was calculated before immunotherapy as follows: platelet count * neutrophil count / lymphocyte count in all included studies, and the cutoff values of the SII ranged from 254.02–2003.95. All studies had an NOS score ≥6. Other information is presented in **Table 1**.

### The association between the SII and PFS in ICI-treated lung cancer patients

Sixteen studies identified a predictive role for the SII in PFS, in patients with lung cancer receiving ICIs. The pooled results demonstrated that an elevated SII was associated with poorer PFS (HR = 1.82, 95% CI: 1.49–2.21; P < 0.001; $I^2$ = 47.3%, P = 0.019) (**Fig 2**). Furthermore, subgroup analysis based on the pathological type (NSCLC: HR = 1.80, 95% CI: 1.45–2.23, P < 0.001; SCLC: HR = 1.99, 95% CI: 1.20–3.29, P = 0.008), line of treatment (first line: HR = 1.55, 95% CI: 1.28–1.89, P < 0.001; second or further line: HR = 2.52, 95% CI: 0.85–7.49, P = 0.097) and combination of other therapies (yes: HR = 1.49, 95% CI: 1.22–1.81, P < 0.001; no: HR = 2.40, 95% CI: 1.79–3.22, P < 0.001) yielded similar results (**Table 2**).

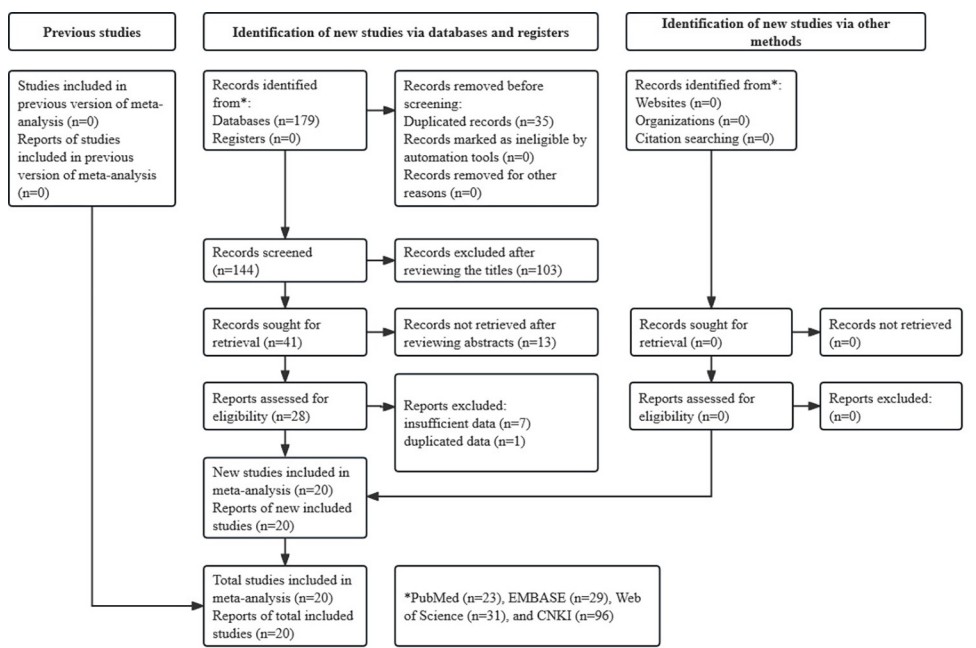

**Fig 1. The flow diagram of this meta-analysis.**

**Table 1. Basic characteristics of included studies.**

| Author | Year | Country | Sample size | Pathological type | Tumor stage | Lines of ICIs | Combination of therapy | Drugs of ICIs | Threshold of SII and determining method | Endpoint | NOS |
|---|---|---|---|---|---|---|---|---|---|---|---|
| Liu [26] | 2019 | China | 44 | NSCLC | TNM IV | ≥2 | None | Nivolumab | 603.5/ROC curve | PFS, OS | 6 |
| Xiong [27] | 2020 | China | 41 | SCLC | Mixed | ≥2 | None | Nivolumab, pembrolizumab, atezolizumab and toripalimab | 730/median value | PFS | 6 |
| Qi [28] | 2021 | China | 53 | SCLC | ES | 1 | Chemotherapy | Atezolizumab | 533.28/ ROC curve | OS | 6 |
| Seban [29] | 2021 | France | 51 | NSCLC | TNM IIIB-IV | 1 | None | Pembrolizumab, | 1270/X-tile software | PFS, OS | 6 |
| Wei [30] | 2021 | China | 64 | NSCLC | TNM IIIB-IV | Mixed | Mixed | Pembrolizumab | 822.39/ ROC curve | PFS | 6 |
| Yang [31] | 2021 | China | 130 | NSCLC | III-IV | Mixed | Mixed | Nivolumab, pembrolizumab, sintilimab, tislelizumab and atezolizumab | 1026/ ROC curve | PFS | 7 |
| Yi [32] | 2021 | China | 121 | NSCLC | TNM IIIB-IV | Mixed | None | Nivolumab, pembrolizumab, sintilimab, tislelizumab, camrelizumab, toripalimab and atezolizumab | 611/ ROC curve | PFS, OS | 7 |
| Banna [33] | 2022 | UK | 308 | NSCLC | TNM IIIB-IV | 1 | Chemotherapy | Not reported | 1444/ ROC curve | PFS, OS | 7 |
| Holtzman [34] | 2022 | Israel | 423 | NSCLC | TNM III-IV | 1 | Mixed | Pembrolizumab | 400/median value | OS | 7 |
| Hu [35] | 2022 | China | 159 | NSCLC | TNM III-IV | Mixed | Mixed | PD-1 inhibitors | 1369.22/ ROC curve | PFS | 6 |
| Liu [36] | 2022 | China | 88 | NSCLC | TNM IIIB-IV | 1 | Chemotherapy and others | Sintilimab | 423/ ROC curve | OS | 7 |
| Xu [37] | 2022 | China | 124 | NSCLC | TNM IIIB-IV | Mixed | None | Nivolumab, pembrolizumab, sintilimab, tislelizumab, camrelizumab and toripalimab | 1146.61/ ROC curve | PFS, OS | 6 |
| Rizzo [38] | 2023 | Italy | 43 | NSCLC | TNM IV | 1 | Mixed | Pembrolizumab | 1235/ ROC curve | PFS, OS | 8 |
| Fang [39] | 2023 | China | 223 | NSCLC | TNM IIIB-IV | 1 | Chemotherapy | PD-1 inhibitors | 792.07/median value | PFS, OS | 6 |
| He [40] | 2023 | China | 58 | NSCLC | TNM IV | Mixed | Chemotherapy | Carrelizumab, Sindilizumab, Tirelizumab and Atezolizumab | 546.5/ ROC curve | OS | 7 |
| Baek [41] | 2024 | Republic of Korea | 55 | SCLC | ES | 1 | Chemotherapy | Not reported | 810/ ROC curve | PFS, OS | 6 |
| Bi [42] | 2024 | China | 178 | NSCLC | TNM III-IV | Mixed | Mixed | Not reported | 2003.95/ ROC curve | PFS, OS | 6 |
| Hua [43] | 2024 | China | 68 | SCLC | ES | 1 | Chemotherapy | Sintilimab, Durvalumab, Tislelizumab, Slunimab, Camrelizumab, Atezolizumab and Envolimab | 254.02/ ROC curve | PFS, OS | 6 |
| Tang [44] | 2024 | China | 92 | NSCLC | TNM IIIB-IV | Mixed | Mixed | Pembrolizumab, nivolumab, camrelizumab, sintilimab, tislelizumab, toripalimab, penpulimab, durvalumab, atezolizumab and sugemalimab | 993.7/ ROC curve | PFS, OS | 6 |

(*Continued*)

**Table 1.** (Continued)

| Author | Year | Country | Sample size | Pathological type | Tumor stage | Lines of ICIs | Combination of therapy | Drugs of ICIs | Threshold of SII and determining method | Endpoint | NOS |
|---|---|---|---|---|---|---|---|---|---|---|---|
| Yamaguchi [45] | 2024 | Japan | 101 | NSCLC | TNM III-IV | Mixed | Mixed | Nivolumab and Ipilimumab | 1160.881/ ROC curve | PFS, OS | 6 |

ICI: immune checkpoint inhibitor; SII: systemic immune-inflammation index; NOS: Newcastle-Ottawa Scale; NSCLC: non-small cell lung cancer; SCLC: small cell lung cancer; TNM: tumor-node-metastasis; PD-1: programmed cell death-1; ROC: receiver operating characteristic curve; PFS: progress-free survival; OS: overall survival.

### The association between SII and OS in ICI-treated lung cancer patients

Sixteen studies identified a predictive role of the SII for OS. The pooled results indicated that an elevated SII was related to worse OS (HR = 2.31, 95% CI: 1.73–3.09, P < 0.001; $I^2$ = 69.7%, P < 0.001) (**Fig 3**). Similarly, subgroup analysis based on pathological type (NSCLC: HR = 2.29, 95% CI: 1.67–3.13, P<0.001; SCLC: HR = 2.65, 95% CI: 1.38–5.07, P = 0.003), line of treatment (first line: HR = 1.74, 95% CI: 1.31–2.30, P<0.001; second or further line: HR = 7.69, 95% CI: 1.94–20.42, P = 0.004) and combination of other therapies (yes: HR = 1.83, 95% CI: 1.30–2.59, P = 0.001; no: HR = 4.87, 95% CI: 1.97–12.03, P = 0.001) produced consistent results (**Table 2**).

### Sensitivity analysis

Sensitivity analysis for PFS and OS indicated that our results were stable and reliable and that none of the included studies showed an impact on the conclusion (**Fig 4A and 4B**).

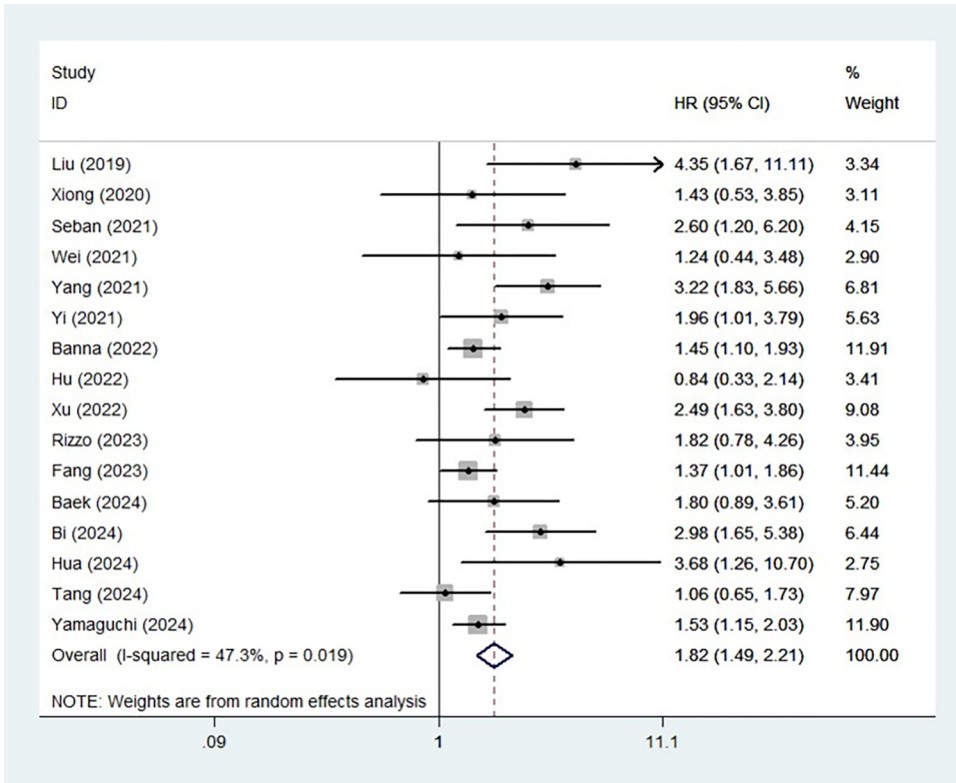

**Fig 2. The association between systemic immune-inflammation index and progression-free survival.**

**Table 2. Results of meta-analysis.**

| Items | No. of studies | Hazard ratio | 95% confidence interval | P value | I$^2$ | P value |
|---|---|---|---|---|---|---|
| **Progression-free survival** | 16 | 1.82 | 1.49–2.21 | <0.001 | 47.3 | 0.019 |
| Pathological type | | | | | | |
| NSCLC | 13 | 1.80 | 1.45–2.23 | <0.001 | 54.3 | 0.010 |
| SCLC | 3 | 1.99 | 1.20–3.29 | 0.008 | 0.0 | 0.412 |
| Lines of treatment | | | | | | |
| First line | 6 | 1.55 | 1.28–1.89 | <0.001 | 3.7 | 0.393 |
| Second or further line | 2 | 2.52 | 0.85–7.49 | 0.097 | 60.4 | 0.112 |
| Combination of other therapies | | | | | | |
| Yes | 4 | 1.49 | 1.22–1.81 | <0.001 | 10.4 | 0.341 |
| No | 5 | 2.40 | 1.79–3.22 | <0.001 | 0 | 0.559 |
| **Overall survival** | 16 | 2.31 | 1.73–3.09 | <0.001 | 69.7 | <0.001 |
| Pathological type | | | | | | |
| NSCLC | 13 | 2.29 | 1.67–3.13 | <0.001 | 72.4 | <0.001 |
| SCLC | 3 | 2.65 | 1.38–5.07 | 0.003 | 29.8 | 0.240 |
| Lines of treatment | | | | | | |
| First line | 9 | 1.74 | 1.31–2.30 | <0.001 | 44.2 | 0.064 |
| Second or further line | 1 | 7.69 | 1.94–20.42 | 0.004 | - | - |
| Combination of other therapies | | | | | | |
| Yes | 7 | 1.83 | 1.30–2.59 | 0.001 | 61.6 | 0.016 |
| No | 4 | 4.87 | 1.97–12.03 | 0.001 | 74.2 | 0.009 |

HR: hazard ratio; NSCLC: non-small cell lung cancer; SCLC: small cell lung cancer.

## Publication bias

The Bess's funnel plots for PFS (Fig 5A) and OS (Fig 5B) were both symmetrical, with Egger's test (P = 0.141; P = 0.108) indicating nonsignificant publication bias.

## Discussion

Our study demonstrates that the SII plays a role in predicting the prognosis of lung cancer patients receiving ICIs, and lung cancer patients with an elevated SII experienced a significantly worse prognosis based on current evidence. Subgroup analysis further confirmed the above findings.

In the last decade, several parameters such as the neutrophil-to-lymphocyte ratio (NLR), platelet-to-lymphocyte ratio (PLR), and lymphocyte-to-monocyte ratio (LMR) have been reported to be related to the outcomes of lung cancer patients treated with ICIs [46–50]. The SII is a novel index based on platelet, neutrophil, and lymphocyte counts. The SII is believed to have greater prognostic value than these indicators in lung cancer, as demonstrated by Liu et al. [26]. The prognostic value of the SII in lung cancer has been determined by several meta-analyses. Zhang et al. included seven studies involving 2786 patients and demonstrated that a high SII was significantly associated with poor OS among lung cancer patients (HR = 1.77, 95% CI: 1.54–2.00, P < 0.001), NSCLC patients (HR = 1.97, 95% CI: 1.69–2.25, P < 0.001), and SCLC patients (HR = 1.38, 95% CI: 1.02–1.85, P < 0.001) [51]. Wang et al. conducted a meta-analysis focusing on NSCLC patients and reported that pretreatment SII was associated with poor OS (HR = 1.88, 95% CI: 1.50–2.36, P < 0.001), disease-free survival (DFS)/PFS (HR = 2.50, 95% CI: 1.20–5.20, P = 0.014), and cancer-specific survival (CSS) (HR = 1.85, 95% CI: 1.19–2.92, P = 0.007) [52]. Moreover, they revealed that, compared with the NLR and PLR,

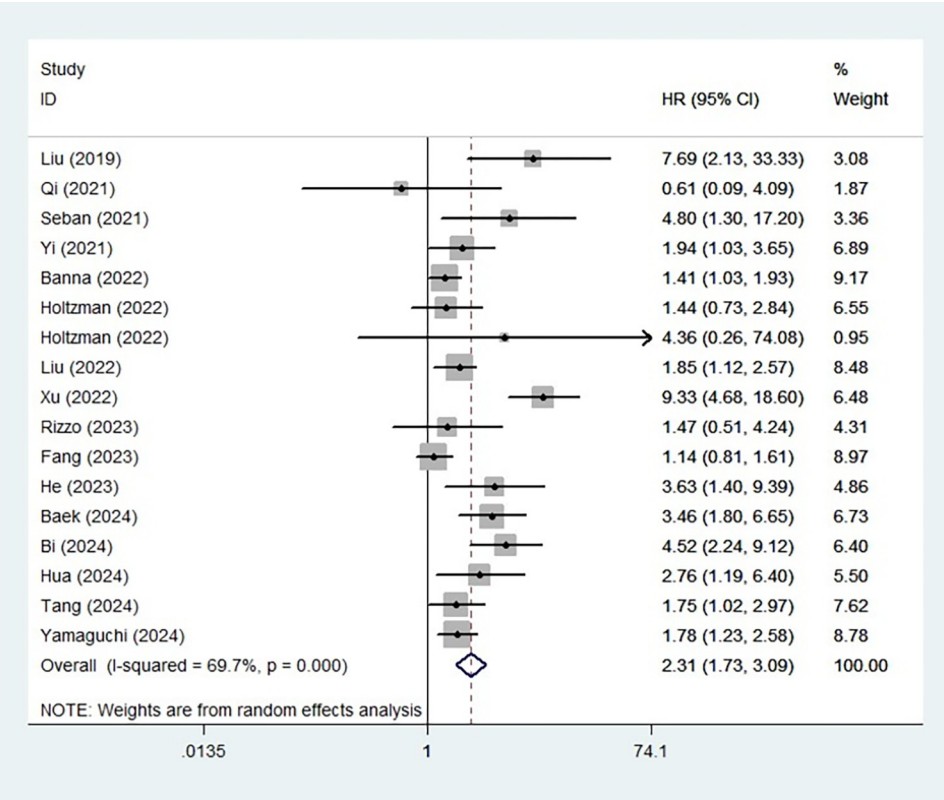

**Fig 3. The association between systemic immune-inflammation index and overall survival.**

the SII had an obviously greater prognostic value in patients with NSCLC [52]. In 2022, a meta-analysis by Zhou et al. included eight studies focusing on SCLC and reported that an elevated SII was related to poor OS (HR = 1.52, 95% CI: 1.15–2.00, P = 0.003) but not PFS (HR = 1.38, 95% CI: 0.81–2.35, P = 0.238) [53]. However, the conclusions of these meta-analyses are relatively rare, and patients receiving ICIs constitute a special group of patients with lung cancer. More specific analyses of the prognostic role of the SII in ICI-treated patients with lung cancer are needed. Therefore, we conducted this study to further characterize the prognostic value of the SII in lung cancer patients treated with ICIs.

Although we demonstrated that the SII can predict PFS and OS in ICI-treated lung cancer patients, the clinical role of the SII in lung cancer patients receiving ICIs is worthy of further

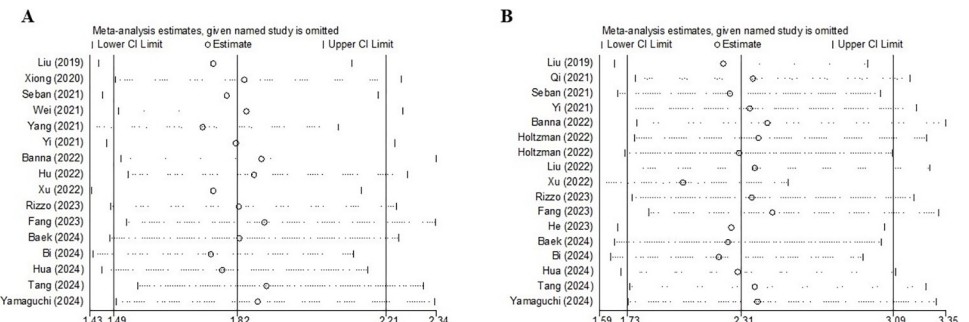

**Fig 4.** Sensitivity analysis for the association between systemic immune-inflammation index and progression-free survival (A) and overall survival (B).

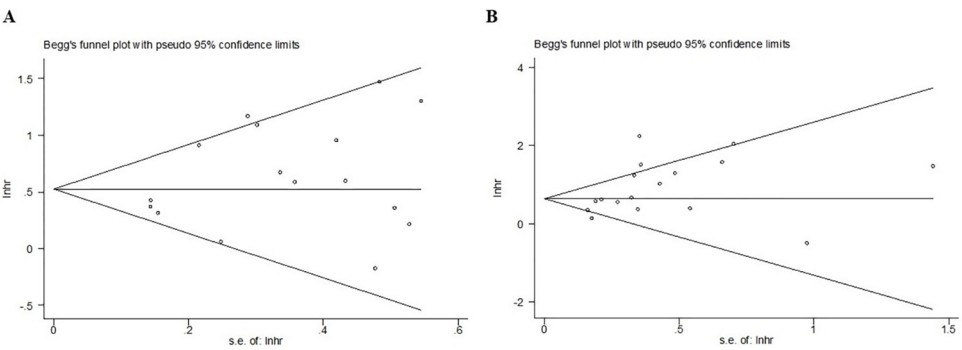

**Fig 5.** Begg's funnel plots for the progression-free survival (A) and overall survival (B).

investigation. For example, among the included studies only three explored the association between the SII and survival of ICI-treated SCLC patients. Thus, the relationship between the SII and the prognosis of patients with SCLC receiving ICIs should be further explored. In addition, our study focused only on the SII. Some studies have revealed that the postimmunotherapy SII may also play a role in predicting long-term survival in patients with lung cancer [27, 54]. Therefore, the prognostic value of the postimmunotherapy SII and changes in the SII during immunotherapy should be further explored. Furthermore, immunotherapy is usually applied in combination with other therapies, such as chemotherapy. Therefore, it is necessary to identify the effect of the combination of other therapies on the prognostic value of the SII in ICI-treated lung cancer patients. Moreover, there are several useful indicators for predicting the efficacy of immunotherapy such as the tumor mutation burden (TMB) and the expression of PD-L1 [55–57]. Thus, a combination of the SII and these parameters might have greater prognostic value.

This meta-analysis had several limitations. First, all included studies were retrospective with relatively small sample sizes, which may have caused bias. Second, most of the included studies were conducted in China, which might affect the generalizability of our conclusions. Third, the cutoff values of the SII among the included studies varied widely and we were unable to determine the optimal cutoff value of the SII in this meta-analysis. Fourth, owing to the lack of original data we could not conduct a subgroup analysis based on other important parameters such as age, pathological subtype, sex, and ICI drugs.

## Conclusion

The SII might serve as a novel and reliable prognostic indicator among lung cancer patients who receive ICIs, and patients with an elevated SII are more likely to have a worse prognosis. More prospective studies are needed to verify the above findings and explore the association between the SII and the prognosis of patients with lung cancer receiving ICIs.

## Supporting information

**S1 Checklist. PRISMA 2020 checklist.**
(DOCX)

**S1 Table. Information about 179 records from databases and reasons of their inclusion or exclusion.**
(DOCX)

## Author Contributions

**Conceptualization:** Yanhui Yang, Xiaoyang Xie.

**Data curation:** Yanhui Yang, Yi Yao.

**Formal analysis:** Yanhui Yang, Lei Luo.

**Investigation:** Ji Li, Lei Luo.

**Methodology:** Ji Li, Lei Luo.

**Resources:** Ji Li.

**Software:** Yi Wang, Yi Yao.

**Supervision:** Xiaoyang Xie.

**Validation:** Yi Wang.

**Writing – original draft:** Yanhui Yang, Yi Wang.

**Writing – review & editing:** Lei Luo, Yi Yao, Xiaoyang Xie.

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
