## [Decision Letter · Decision Letter 0]

20 Sep 2024

PONE-D-24-32840Prognostic value of pretreatment systemic immune-inflammation index in lung cancer patients receiving immune checkpoint inhibitorsPLOS ONE

Dear Dr. Xie,

Thank you for submitting your manuscript to PLOS ONE. After careful consideration, we feel that it has merit but does not fully meet PLOS ONE’s publication criteria as it currently stands. Therefore, we invite you to submit a revised version of the manuscript that addresses the points raised during the review process.

Minor revisions have been proposed.  Please address these before resubmission.

We look forward to receiving your revised manuscript.

Kind regards,

Gayle E. Woloschak, PhD

Section Editor

PLOS ONE

Journal requirements: 1. When submitting your revision, we need you to address these additional requirements. Please ensure that your manuscript meets PLOS ONE's style requirements, including those for file naming. The PLOS ONE style templates can be found at https://journals.plos.org/plosone/s/file?id=wjVg/PLOSOne_formatting_sample_main_body.pdf and https://journals.plos.org/plosone/s/file?id=ba62/PLOSOne_formatting_sample_title_authors_affiliations.pdf. 2. We noticed you have some minor occurrence of overlapping text with the following previous publication(s), which needs to be addressed: 10.7759/cureus.52720 In your revision ensure you cite all your sources (including your own works), and quote or rephrase any duplicated text outside the methods section. Further consideration is dependent on these concerns being addressed. 3. We note that the grant information you provided in the ‘Funding Information’ and ‘Financial Disclosure’ sections do not match.  When you resubmit, please ensure that you provide the correct grant numbers for the awards you received for your study in the ‘Funding Information’ section. 4. As required by our policy on Data Availability, please ensure your manuscript or supplementary information includes the following:  A numbered table of all studies identified in the literature search, including those that were excluded from the analyses.   For every excluded study, the table should list the reason(s) for exclusion.   If any of the included studies are unpublished, include a link (URL) to the primary source or detailed information about how the content can be accessed.  A table of all data extracted from the primary research sources for the systematic review and/or meta-analysis. The table must include the following information for each study:  Name of data extractors and date of data extraction  Confirmation that the study was eligible to be included in the review.   All data extracted from each study for the reported systematic review and/or meta-analysis that would be needed to replicate your analyses.  If data or supporting information were obtained from another source (e.g. correspondence with the author of the original research article), please provide the source of data and dates on which the data/information were obtained by your research group.  If applicable for your analysis, a table showing the completed risk of bias and quality/certainty assessments for each study or outcome.  Please ensure this is provided for each domain or parameter assessed. For example, if you used the Cochrane risk-of-bias tool for randomized trials, provide answers to each of the signalling questions for each study. If you used GRADE to assess certainty of evidence, provide judgements about each of the quality of evidence factor. This should be provided for each outcome.   An explanation of how missing data were handled.   This information can be included in the main text, supplementary information, or relevant data repository. Please note that providing these underlying data is a requirement for publication in this journal, and if these data are not provided your manuscript might be rejected.  

Additional Editor Comments:

Both reviewers have suggested minor revisions for this work.

Reviewers' comments:

Reviewer's Responses to Questions

**Comments to the Author**

1. Is the manuscript technically sound, and do the data support the conclusions?

Reviewer #1: Yes

Reviewer #2: Yes

2. Has the statistical analysis been performed appropriately and rigorously? 

Reviewer #1: Yes

Reviewer #2: Yes

3. Have the authors made all data underlying the findings in their manuscript fully available?

Reviewer #1: Yes

Reviewer #2: Yes

4. Is the manuscript presented in an intelligible fashion and written in standard English?

Reviewer #1: Yes

Reviewer #2: Yes

5. Review Comments to the Author

Reviewer #1: Thanks for giving me the opportunity to review this interesting work. In overall, this manuscript was well designed and organized. I have only a few minor concerns about the work.

1. Some grammar and spelling mistakes could be found in the manuscript.

2. This work is a meta-analysis. In general, the study design should be noted in the title.

3. Databases searched should be mentioned in the abstract.

4. Although the authors indicated in the inclusion criteria that SII was calculated before immunotherapy, I would suggest the authors deleting the term "pretreatment" which might lead to ambiguity. Patients may receive other therapies before the immunotherapy?

Reviewer #2: The authors eplored the prognostic role of SII in ICIs treated patients. Congratulations for this successful work. In overall, this study was well designed and written. I have several minor concerns about this paper.

1. Study design type should be specificly described in the title.

2. "patients were divided into elevated and normal-SII groups according" This sentence is not correct?

3. The results of quality assessment should be also described in the results part.

4. Besides, the authors should carefully check and revise the manuscript for grammar or spelling errors.

6. PLOS authors have the option to publish the peer review history of their article (what does this mean?). If published, this will include your full peer review and any attached files.

Reviewer #1: No

Reviewer #2: No

---

## [Author Response · Author response to Decision Letter 0]

2 Oct 2024

Response to issues by editor:

Question 1: Grant information and Financial Disclosure

Answer 1: We have changed the grand information as follows: “Funding: None”.

Question 2: Data sharing statement

Answer 2: This is a meta-analysis, all the data used have been presented in the manuscript. Therefore, the sharing for raw data is not applicable in our manuscript.

Question 3: PRISMA checklist

Answer 3: The checklist has been uploaded.

Response to editor:

Question 1. When submitting your revision, we need you to address these additional requirements.

Answer 1: Thanks for your comment. We have carefully modified our manuscript according to the guidelines as request.

Question 2. We noticed you have some minor occurrence of overlapping text with the following previous publication(s), which needs to be addressed: 

10.7759/cureus.52720

In your revision ensure you cite all your sources (including your own works), and quote or rephrase any duplicated text outside the methods section. Further consideration is dependent on these concerns being addressed.

Answer 2: We have carefully modified our manuscript to reduce the overlapping text.

Question 3. We note that the grant information you provided in the ‘Funding Information’ and ‘Financial Disclosure’ sections do not match. 

Answer 3: We have updated the funding information and financial disclosure part in the submission system.

Response to Reviewer #1: 

Thanks for giving me the opportunity to review this interesting work. In overall, this manuscript was well designed and organized. I have only a few minor concerns about the work.

Question 1. Some grammar and spelling mistakes could be found in the manuscript.

Answer 1: This manuscript has been edited by AJE service with the verification code 5FF8-A3F4-9204-0BA6-ACEB.

Question 2. This work is a meta-analysis. In general, the study design should be noted in the title.

Answer 2: We have revised the title to “Prognostic value of systemic immune-inflammation index in lung cancer patients receiving immune checkpoint inhibitors: a meta-analysis”

Question 3. Databases searched should be mentioned in the abstract.

Answer 3: Databases searched has been added in the abstract. “PubMed, EMBASE, Web of Science, and CNKI databases”.

Question 4. Although the authors indicated in the inclusion criteria that SII was calculated before immunotherapy, I would suggest the authors deleting the term "pretreatment" which might lead to ambiguity. Patients may receive other therapies before the immunotherapy?

Answer 4: Thanks for your question. After careful consideration, we have deleted the term of “pretreatment”.

Response to Reviewer #2: The authors eplored the prognostic role of SII in ICIs treated patients. Congratulations for this successful work. In overall, this study was well designed and written. I have several minor concerns about this paper.

Question 1. Study design type should be specificly described in the title.

Answer 1: We have revised the title to “Prognostic value of systemic immune-inflammation index in lung cancer patients receiving immune checkpoint inhibitors: a meta-analysis”

Question 2. "patients were divided into elevated and normal-SII groups according" This sentence is not correct?

Answer 2: This sentence has been modified.

Question 3. The results of quality assessment should be also described in the results part.

Answer 3: We have added the description about the quality assessment results. “All studies were with a NOS score≥6.”

Question 4. Besides, the authors should carefully check and revise the manuscript for grammar or spelling errors.

Answer 4: This manuscript has been edited by AJE service with the verification code 5FF8-A3F4-9204-0BA6-ACEB.

---

## [Decision Letter · Decision Letter 1]

10 Oct 2024

Prognostic value of the systemic immune-inflammation index in lung cancer patients receiving immune checkpoint inhibitors: a meta-analysis

PONE-D-24-32840R1

Dear Dr. Xie:

We’re pleased to inform you that your manuscript has been judged scientifically suitable for publication and will be formally accepted for publication once it meets all outstanding technical requirements.

Kind regards,

Gayle E. Woloschak, PhD

Section Editor

PLOS ONE

Additional Editor Comments (optional):

Both reviewers agreed to accept this work.

Reviewers' comments:

Reviewer's Responses to Questions

**Comments to the Author**

1. If the authors have adequately addressed your comments raised in a previous round of review and you feel that this manuscript is now acceptable for publication, you may indicate that here to bypass the “Comments to the Author” section, enter your conflict of interest statement in the “Confidential to Editor” section, and submit your "Accept" recommendation.

Reviewer #1: All comments have been addressed

Reviewer #2: All comments have been addressed

2. Is the manuscript technically sound, and do the data support the conclusions?

Reviewer #1: Yes

Reviewer #2: Yes

3. Has the statistical analysis been performed appropriately and rigorously? 

Reviewer #1: Yes

Reviewer #2: Yes

4. Have the authors made all data underlying the findings in their manuscript fully available?

Reviewer #1: Yes

Reviewer #2: Yes

5. Is the manuscript presented in an intelligible fashion and written in standard English?

Reviewer #1: Yes

Reviewer #2: Yes

6. Review Comments to the Author

Reviewer #1: This work identified the association between the systemic immune-inflammation index (SII) score and prognosis in immune checkpoint inhibitor (ICI)-treated patients with lung cancer. I have no more comments on this paper now.

Reviewer #2: The authors have carefully revised their manuscript as request. I would suggest the publication for this paper.

7. PLOS authors have the option to publish the peer review history of their article (what does this mean?). If published, this will include your full peer review and any attached files.

Reviewer #1: No

Reviewer #2: No

---

## [Editor Report · Acceptance letter]

23 Oct 2024

PONE-D-24-32840R1 

PLOS ONE

Dear Dr. Xie, 

I'm pleased to inform you that your manuscript has been deemed suitable for publication in PLOS ONE. Congratulations! Your manuscript is now being handed over to our production team.

Kind regards, 

on behalf of

Dr. Gayle E. Woloschak 

Section Editor

PLOS ONE